# Drought reduces blue-water fluxes more strongly than green-water fluxes in Europe

René Orth[1,2] & Georgia Destouni [1]

Drought comprehensively affects different interlinked aspects of the terrestrial water cycle, which have so far been mostly investigated without direct comparison. Resolving the partitioning of water deficit during drought into blue-water runoff and green-water evapotranspiration fluxes is critical, as anomalies in these fluxes threaten different associated societal sectors and ecosystems. Here, we analyze the propagation of drought-inducing precipitation deficits through soil moisture reductions to their impacts on blue and green-water fluxes by use of comprehensive multi-decadal data from > 400 near-natural catchments along a steep climate gradient across Europe. We show that soil-moisture drought reduces runoff stronger and faster than it reduces evapotranspiration over the entire continent. While runoff responds within weeks, evapotranspiration can be unaffected for months. Understanding these drought-impact pathways across blue and green-water fluxes and geospheres is essential for ensuring food and water security, and developing early-warning and adaptation systems in support of society and ecosystems.

[1] Department of Physical Geography, Bolin Centre for Climate Research, Stockholm University, SE-10691 Stockholm, Sweden. [2] Department of Biogeochemical Integration, Max Planck Institute for Biogeochemistry, D-07745 Jena, Germany. Correspondence and requests for materials should be addressed to R.O. (email: rene.orth@bgc-jena.mpg.de)

Drought impacts are multifaceted[1,2] and can affect water fluxes in the hydrosphere[3] (blue-water runoff[4]) and/or the biosphere[5,6] (green-water evapotranspiration, ET[4]), with serious societal impacts, e.g., on water security[7], food security[8], infrastructure[9] and health[10]. For example, since 1980, droughts across the United States have led to losses of > 200 billion USD and almost 3000 fatalities (https://www.ncdc.noaa.gov/billions/summary-stats).

Precipitation deficits are a recognized main driver of drought[1,2], leading to soil-moisture decrease[11] and further to water-deficit propagation and partitioning between ET and runoff fluxes in the landscape. Moreover, ET increase driven by climate change[1] and/or by human land- and water-use changes in the landscape[12,13] can also intensify droughts[14].

However, the partitioning of drought-related water deficits between green (ET) and blue-water (runoff) water fluxes remains largely unresolved, especially on large scales and under different/changing climatic conditions[2,15]. Yet, this partitioning is key to understanding and developing appropriate mitigation-adaptation strategies for the various possible societal and ecosystem impacts of droughts. Specifically, green-water flux anomalies are primarily associated with vegetation impacts, including on agricultural crops[16] and forestry[17], whereas blue-water anomalies affect, e.g., energy[18], dam[16], and freshwater[19] security, as well as irrigation capacity[20], thus further adding to agricultural drought impacts. Insufficient understanding and quantification of the drought-impact partitioning between blue and green-water fluxes, and of the variation of this partitioning under different climate conditions may, therefore, seriously mislead or impede relevant societal responses to drought and its possible future intensification.

In this study, we explore and compare the development of soil moisture droughts and their impact evolution across geospheres into changes of blue and green-water fluxes. We show that drought-inducing precipitation deficits propagate through soil moisture droughts to reduce blue-water runoff (impact the hydrosphere) stronger and faster than they reduce green-water evapotranspiration (impact the biosphere) across all European climate regimes.

## Results

**Approaching drought in a multivariate framework.** In this study, we focus on soil-moisture drought periods. Therein, we analyze the propagation of precipitation deficits through soil moisture decreases to associated changes in runoff and ET fluxes across geospheres by use of comprehensive multi-decadal data from > 400 near-natural European catchments across three regions characterized by different climate regimes (Fig. 1). Climate in this study is determined through the dryness index[21], i.e., the location-specific ratio between the long-term averages of annual net radiation (unit-adjusted) and precipitation. By considering relatively undisturbed landscapes, the study focuses on natural, climate-related rather than anthropogenic, landscape-related drought drivers.

Analyzing continental water fluxes from an observational perspective, as is the aim of this study, is necessarily a compromise between using sparse station observations and model-based estimates that provide full spatial and temporal coverage. Addressing this issue, we employ runoff measurements from catchments distributed across Europe (ref. [22], see methods) and gridded precipitation data derived by upscaling station observations[23], along with gridded reanalysis-like data on ET[24] and soil moisture[25]. While the latter products are model-based, they are validated against corresponding station observations[24,25]; to further check their usefulness for our analysis we compare them against more station observations, and repeat the analysis

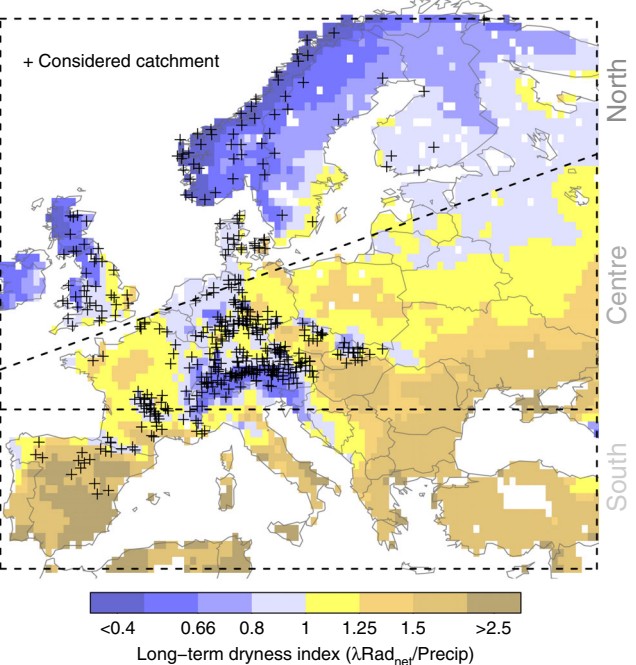

**Fig. 1** Study locations and their climate. Considered European climate regions, defined as in ref. [1], and catchments therein. Colors denote climate regime as determined by dryness index[21]

by replacing them with similar datasets obtained with different models to ensure negligible influence of particular models on our conclusions (see below).

Overall, the employed data products are largely independent. Each investigated catchment is defined by its corresponding runoff-observation location in the landscape, and most catchments are smaller than 1000 km². For a catchment-based comparison of all quantities, each catchment is characterized by the associated observed runoff and by the corresponding gridded data for the 0.5° x 0.5° grid-cell in which (most of) the catchment is located. Further, all considered fluxes are normalized by catchment area to units of mm/day prior to the analysis.

In order to comprehensively investigate the partitioning of drought-related water deficits between blue and green-water fluxes, we employ a drought composite analysis approach[26]. We quantify drought in terms of soil moisture anomalies, a simple and relevant drought index[1,26]. Anomalies in this study are computed by removing the mean seasonal cycle from the actual time series for each catchment. We then analyze composites of the ten strongest soil-moisture droughts (greatest reductions from seasonal mean soil moisture at each time) occurring in each catchment during the 24-year period 1984–2007. While there are numerous ways to quantify drought strength[1,2] we focus on the magnitude of the maximum soil moisture anomaly. For this purpose, we first identify the driest half-monthly, total-column soil-moisture anomaly in the warm season (May–September) of each year, yielding 24 annual driest anomalies. From these, we select the ten strongest anomalies, and finally we determine the mean drought period as the total time before drought peak (drought-buildup period) and after drought peak (recovery period), over which soil moisture is reduced below normal in the majority of the selected ten drought years. Soil-moisture levels are, therefore, close to normal (i.e., unchanged by drought and thus zero anomaly) at the beginning and end of each drought period, such that changes in soil-water storage beyond the mean

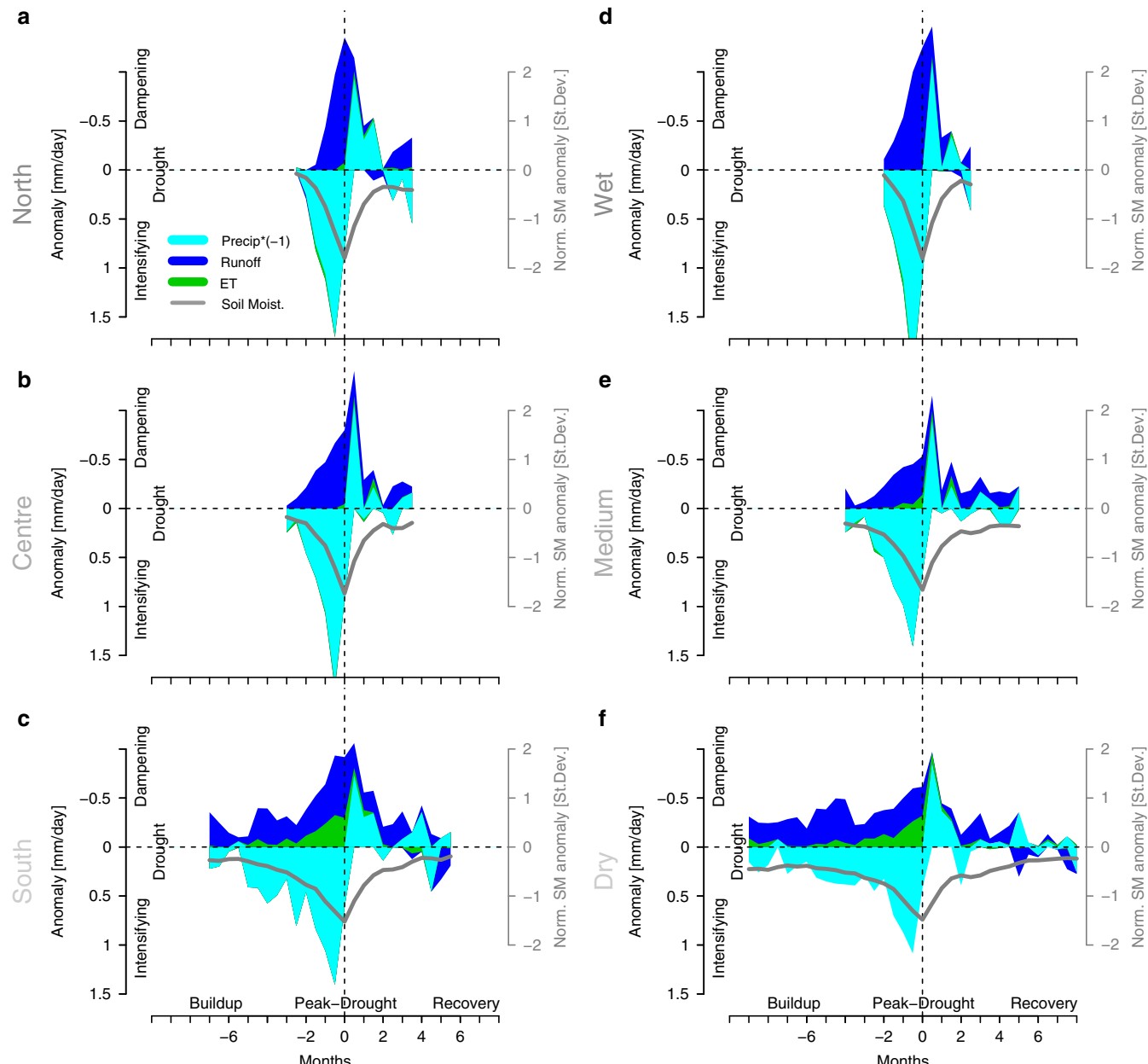

**Fig. 2** Evolution of major water balance components during droughts. Time evolution of anomalies in precipitation, runoff, and evapotranspiration (ET) averaged across the ten strongest annual droughts during 1984–2007. **a–c** Mean evolution over all catchments in each region. **d–f** Mean evolution over catchments grouped into three equally large groups with respect to dryness index. Further, the corresponding mean soil moisture evolution during drought is shown in gray with the y-axis on the right

seasonal variations over the period can be neglected. Using precipitation, runoff, and ET data for these mean drought periods, we compute composites across all considered droughts to study temporal drought-evolution (as also done in ref. [26]). Moving beyond the drought composite analysis in ref. [26], we furthermore quantify accumulated anomalies of precipitation, runoff and ET, and refer to these in following as the drought-related anomalies of each flux variable.

**Temporal drought and flux evolution.** Figure 2 shows the mean temporal evolution of precipitation, runoff, and ET anomalies during drought, with the anomalies grouped in terms of whether they intensify or dampen drought, instead of their actual sign.

Results are shown as averages across all catchments in each of the three European regions (Fig. 2a–c), and as averages across all catchments located in comparable climate (i.e., dryness) conditions (Fig. 2d–f). The similarity of the results in Fig. 2a–c and 2d–f indicates a dominant role of the dryness of different climates across the investigated European sub-regions in shaping the temporal drought response. Remarkable runoff reductions are generally found shortly after drought onset, while ET decreases only occur after several months and solely in the dry southern Europe. This shows that drought impacts are faster on blue than on green-water fluxes. Overall, in the investigated near-natural catchments and climate regimes, soil-moisture droughts are almost exclusively induced by below-normal precipitation, with ET increases playing virtually no role in initiation of the

investigated droughts. The strongest anomalies are observed just before the peak of the drought. After drought peak, the soil-moisture deficit is recovered by above-normal precipitation and continued below-normal runoff, with nearly no post-peak ET anomaly exhibited for any climate.

By fundamental water balance, any net imbalance of input (precipitation) and output (ET, runoff) fluxes occurring at any point in time during the drought period must correspond to an associated change in water storage within the catchment at that time (assuming negligible roles of measurement uncertainties and groundwater flow over catchment boundaries). The generally strong and persistent drought reduction of blue-water flux (runoff) counteracts then water storage depletion, while the relatively unaffected green-water flux (ET) promotes it. Interestingly, the continued dry runoff anomaly after drought peak is opposite to the concurrent wet precipitation anomaly. This illustrates a blocking of the water cycle. Similar non-linear rainfall-runoff behavior has been found in previous studies[27,28]. The blocking is caused by the excess precipitation water in this drought phase largely contributing to refill the subsurface water storage, without substantially adding to runoff. As soil moisture increases and the groundwater table is raised in this process, also groundwater runoff must increase due to higher saturated than unsaturated hydraulic conductivity of soils combined with increased hydraulic gradient towards surface water.

Similar results to Fig. 2 are obtained across catchments of different size as shown in Supplementary Fig. 1 for the smallest (size < 50 km²) and largest (size > 1000 km²) considered catchments, even though the runoff response is slightly faster for the small catchments. While the runoff and precipitation data underlying these results are (based on) observations, the soil moisture and ET data are (commonly used) model products. Nevertheless, the ET findings are deemed reliable based on agreement of drought-related ET anomalies from the employed dataset with Fluxnet data (ground truth, http://fluxnet.fluxdata.org/data/fluxnet2015-dataset/) (Supplementary Fig. 2). Furthermore, there is significant agreement between the employed soil moisture product and independent measurements across the considered climate regimes (Supplementary Fig. 3). This suggests that drought periods are correctly captured by the soil moisture product, which is further indicated by the consistent observed runoff and precipitation responses shown in Fig. 2. Finally, to test the spatial representativeness of our catchment sampling (which is unavoidable in oder to include observed runoff in the analysis). Figure 2 is also re-computed with gridded data for the whole European regions (Supplementary Fig. 4). Similar results are obtained, indicating robustness of the catchment-based findings.

The relatively fast propagation of drought impacts through the terrestrial hydrosphere to blue-water runoff can be mechanistically explained by a lower than normal groundwater table being associated with dry soil-moisture anomalies;[14] this implies a decreased hydraulic gradient and reduced flux of groundwater into downgradient streams, thereby leading to decreased resulting runoff. Furthermore, considerably longer drought duration in the dry climate of southern Europe than in the wetter climates of central and northern Europe (Fig. 2) can explain why drought impacts on ET are limited to dry climate. Increased radiation, which commonly occurs in droughts (Supplementary Fig. 4), compensates for the negative impact of dry (root-zone) soil moisture on vegetation and hence on ET during the shorter droughts of relatively wet climates. For the more prolonged droughts of dry climate, however, the upward flux rate of soil-water required to maintain vegetation activity and associated ET cannot be sustained, as both plant-available water and hydraulic conductivity decrease greatly with increasingly unsaturated soil conditions.

The water limitations that determine the changes in runoff and ET during drought-evolution are thus of different nature. This is also supported by runoff anomalies correlating similarly well with both soil moisture and precipitation anomalies, whereas ET anomalies correlate primarily with soil moisture anomalies (Supplementary Fig. 5). Runoff reductions are explained by any precipitation deficit limiting the water amount available for runoff, and any soil-moisture deficit controlling the groundwater hydraulic gradient and associated groundwater flow feeding the runoff. These drought-related water limitations affect runoff across all European climates. In contrast, ET reductions are explained by soil moisture deficits mostly in the dry climate of southern Europe where prolonged soil moisture deficits limit both the plant-available water and the unsaturated hydraulic conductivity, and thus the upward flux of soil-water that can feed the plants.

**Water flux changes in different droughts and climates.** For an assessment of changes in blue and green-water fluxes under different drought strengths, Fig. 3 displays accumulated anomalies of runoff and ET, and of vegetation activity and drought-buildup duration, for different drought strengths and climate regimes. Drought strength is here expressed in terms of accumulated precipitation deficit, and climate regime is determined in terms of dryness index (see Fig. 1). Runoff is generally reduced during droughts of different strengths, and runoff anomalies tend to be increasingly negative for greater strengths (precipitation deficits). Runoff reductions are further similar across dry (dryness index > 1), intermediate (dryness index ≈ 1) and wet (dryness index < 1) climate regimes (Fig. 3a; see also the geographical distribution of runoff anomalies in Supplementary Fig. 6a and the point-by-point relation to dryness index and accumulated precipitation deficit in Supplementary Fig. 6b). In contrast, the ET responses to droughts of different strength depend to a large degree on the location-specific long-term dryness index, with ET reductions appearing only in dry climate, and even (small) ET increases occurring in wet climate (Fig. 3b; see also the geographical distribution of ET anomaly in Supplementary Fig. 6c and the point-by-point relation to dryness index and accumulated precipitation deficit in Supplementary Fig. 6d).

While most of the ET anomaly variability is explained by different dryness indices and precipitation deficits (Supplementary Fig. 6d), this is not the case for runoff (Supplementary Fig. 6b). The remaining runoff variability may depend on different aquifer characteristics, yielding different groundwater storage and flow changes and thereby different runoff responses to drought in different catchments. Even though such aquifer variability can lead to locally different drought responses in blue vs. green-water fluxes, Fig. 3 shows that blue-water runoff is generally more strongly affected by drought than green-water ET in all considered climate regimes and for droughts of all investigated magnitudes. The latter further indicates that the (necessarily) arbitrary selection of the ten strongest droughts in the above drought-evolution part of this study should not greatly impact the related results and conclusions.

The ET responses to droughts of different strength are largely consistent with those of vegetation activity (Fig. 3c). This underlines an important role of plant transpiration for ET across Europe[29], even though the contribution of transpiration to ET varies across time and space, and is still under debate[30]. The drought-induced ET and NDVI increases in humid areas may seem counter-intuitive; however, they can be explained as droughts are not only characterized by reduced water availability, but also by increased radiation and hence energy availability (Supplementary Fig. 4). In that sense, drought in wet and,

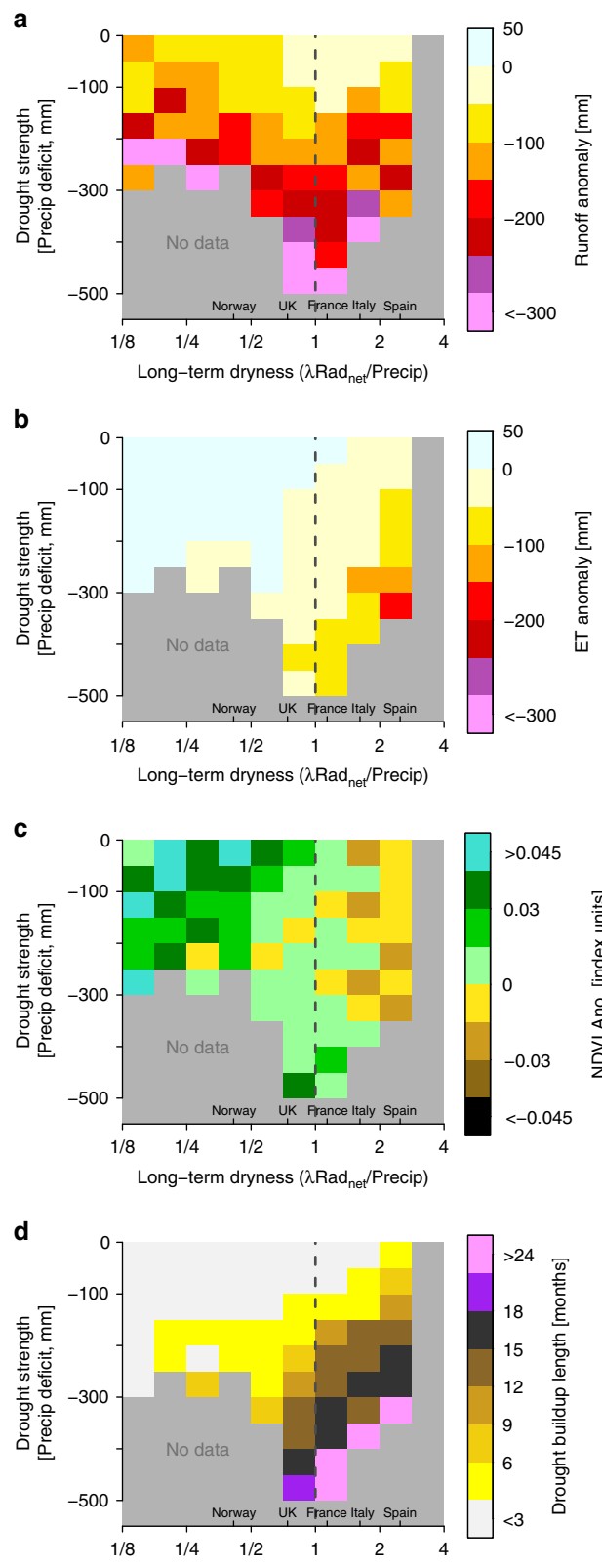

**Fig. 3** Contrasting drought response of runoff, ET, and vegetation. Accumulated anomalies of **a** runoff, **b** ET, **c** Normalized Difference Vegetation Index (NDVI) during drought-buildup, and **d** length of the drought-buildup period, for different drought strengths (in terms of accumulated precipitation deficit, y-axis) and climate regimes (in terms of long-term dryness index, x-axis). Values shown in each box are means across all droughts characterized by the respective precipitation deficit and occurring in catchments with respective long-term dryness

limited climate regimes with long droughts, such as in Spain (Fig. 3d), the above-average radiation is insufficient for counter-acting the negative vegetation impacts and associated ET decreases. Differences seen between ET and vegetation responses associated with the greatest precipitation deficits in intermediate climate (Fig. 3b, c, respectively) may be due to human drought interventions (e.g., irrigation) in countries like France and the United Kingdom; such intervention effects can only be captured by satellite-based vegetation activity data and not by model-based ET data. Further relevant in terms of drought impacts is that the greatest precipitation water deficits are found in such inter-mediate climate; smaller deficits in drier and wetter climates are due to their lower normal precipitation and shorter dry spells, respectively. Moreover, drought duration is related to drought strength expressed through the associated precipitation deficit (Fig. 3d). Such a relationship across different measures of drought strength suggests that our conclusions are insensitive to choices of such strength measures for characterizing droughts.

Figure 4 further compares the absolute water amounts associated with drought-related anomalies in cumulative pre-cipitation, runoff, and ET fluxes, averaged across the catchments in the three European regions. As in Fig. 2, anomalies are not shown with their actual sign, but grouped in terms of whether they build up or recover drought. The mean drought-inducing precipitation deficits increase from the cold and humid northern climate (~60 mm) towards the warm and dry climate in the South (~110 mm). A salient result across all regions is the dominant role of runoff anomalies in compensating the precipitation-induced water deficits; runoff decreases account for 65–80% of the precipitation deficits. In comparison, ET reductions are small and, consistently with the results above, only notable in the dry climate regime (accounting for 0–20% of the precipitation deficits).

Similar results to Fig. 4 are also obtained with alternative gridded data on precipitation, ET and soil moisture (Supplementary Fig. 7). These general results are also consistent with site- and time-specific results of an earlier study, investigating the 2003 summer at the Swiss Rietholzbach site[31]. This highlights the robustness of our findings as they are valid across independent datasets. The uncertainties associated with all water quantities (error bars in Fig. 4) highlight that drought responses are variable across catchments and depend on local properties, such as soil and vegetation type, and terrain and aquifer characteristics. To reveal typical drought response patterns for different climatic conditions, as done in this study, it is, therefore, essential to spatially aggregate results across many catchments, representing a variety of local conditions.

**Drought propagation across geospheres**. Summarizing the main results of this study, Fig. 5 illustrates the drought propagation across geospheres. Drought is induced by the atmosphere, i.e., by an accumulating precipitation water deficit, which propagates almost directly into the soil-water in the terrestrial hydrosphere, causing soil moisture decrease within days across Europe. The

therefore, energy-limited climate regimes (e.g., Norway) promotes biospheric activity, as seen from increases in ET and NDVI, because this is primarily controlled by (increased) net radiation. The dominance of this effect is possible thanks to the relatively short drought duration in wet climate (Fig. 3d). In dry and water-

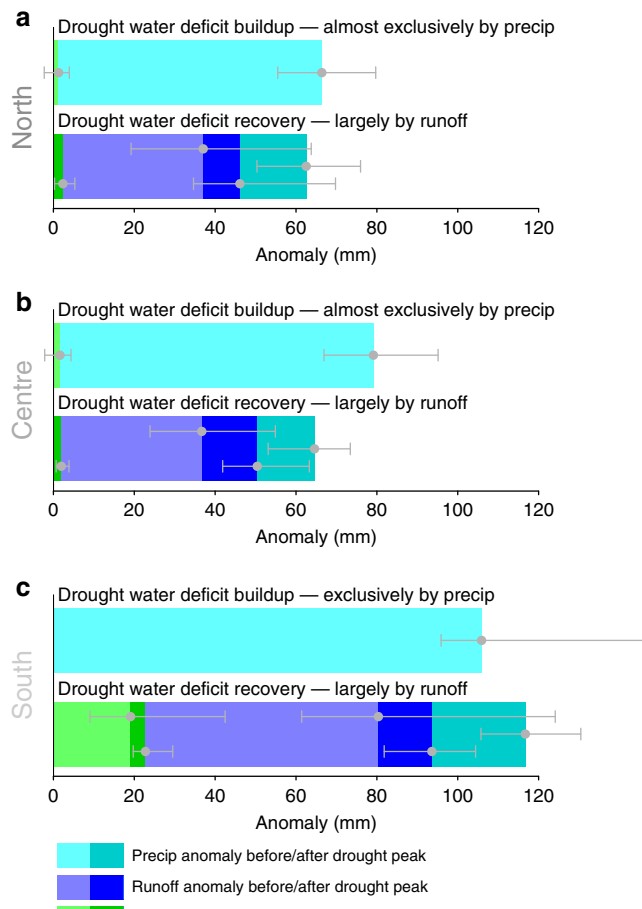

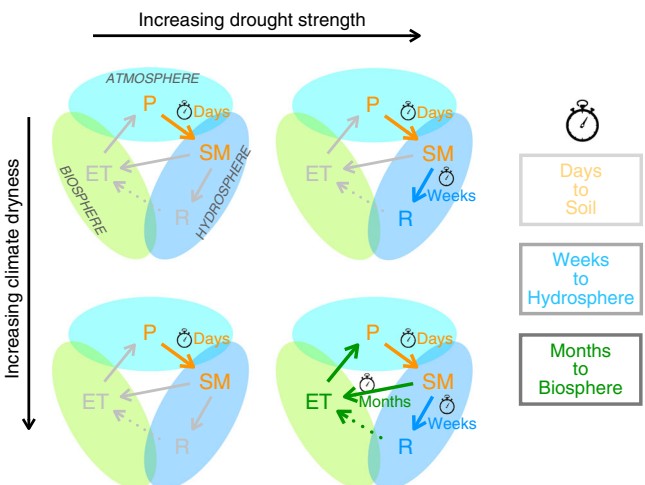

**Fig. 5** When and where drought jumps geospheres. Propagation of drought-inducing precipitation (P) deficits to soil moisture change (SM), through the whole catchment hydrosphere to blue-water runoff (R), and to the biosphere through green-water evapotranspiration (ET) back into atmosphere. Weaker droughts only affect soil moisture change, while the impacts of stronger droughts propagate through the entire hydrosphere to R in all climates, and to ET and the biosphere in dry climate. This summary is valid across the investigated European climate regimes and drought strengths

**Fig. 4** Comparing drought-related precipitation, runoff, and evapotranspiration deficits. **a-c** Mean drought-related behavior of the major water-balance fluxes (precipitation, runoff, ET) averaged across the ten strongest annual droughts and over all catchments in each European climate region. Error bars denote the inter-quantile range (25th–75th percentile) across results from all catchments in the respective region

soil-moisture decrease further partitions and propagates to reduction of blue-water runoff within weeks across all climates, while it hits green-water ET and associated vegetation in the biosphere much later, within months, and then only in the dry climate of southern Europe. This key finding represents a spatio-temporal paradox: blue-water runoff is impacted faster than green-water ET even though the length scales of the water pathways transmitting the drought-induced deficits are much longer for runoff than for ET (see schematic illustration in Supplementary Fig. 8). Specifically, the ET pathways are essentially vertical through the root-zone soil and plants, with lengths on the order of $10^0$ m, while the runoff pathways have variable lengths on the order of $10^1$–$10^4$ m, going first essentially vertically through the soil-water and then laterally through groundwater to and through the stream network until the runoff observation point of each catchment.

Moreover, the longer a precipitation deficit persists and the further drought consequently propagates across geospheres, the longer it takes for the hydrosphere water to recover to its normal magnitude (in soil moisture and runoff, Figs. 2 and 3). In contrast, the ET and associated plant transpiration of the biosphere recover almost immediately after drought peak. It remains to be further investigated if the latter behavior would also occur in drier climate regimes outside Europe.

## Discussion

The presented results suggest that drought response measures need to be tailored to prevailing climate regime and elapsed drought duration. In particular, in wetter climate and/or early into a drought, response measures should focus on adapting to lower runoff levels, e.g., by adjusting dam operations for increased support of downstream water uses, navigation, and aquatic eco-systems. In drier climate, and/or further into a drought, the focus should be extended or even shifted to targeted irrigation support of essential crops and vegetation, while balancing and temporarily limiting other water uses, and also preparing communities for higher temperatures induced by lower ET and consequently increased sensible heat flux.

This study is a pioneer effort in joint analysis and direct quantitative comparison of large-scale drought impacts on land-atmosphere interactions as well as on water resource conditions in the landscape. While the study focus has been on Europe and other regions remain yet to be investigated, the present findings should trigger more exchanges and joint efforts of different geosphere science communities. Highlighting the potential of such interdisciplinarity, this investigation has advanced the understanding of drought-impact evolution across geospheres, and revealed essential early-warning and mitigation-adaptation potential, in that blue and green-water fluxes are impacted weeks or months after drought initiation, respectively. These response times can be exploited in combination with hydro-meteorological forecasting and operational drought monitoring (http://droughtmonitor.unl.edu), paving the way for improved management of freshwater resources under forthcoming climate change with possible increasing drought frequency and/or magnitude[1]. Furthermore, for land-atmosphere interactions, the study has highlighted contrasting drought responses in dry vs. wet climate, calling for further investigation and testing in other parts of the world.

## Methods

**Temporal and spatial aggregation of data**. All data used in this study has been aggregated to half-monthly periods, except for the vegetation activity data for

which this is the native temporal resolution, and which, therefore, is limiting to the temporal resolution of all analyses performed in this study. The aggregation was done by separately computing the means for the first 15 days of each month, and of the remaining days. Mean values were computed if at least 6 days of data were available in a half-monthly period. All employed gridded data products cover (at least) Europe and have native spatial resolutions of 0.5° x 0.5° or higher. Datasets with higher resolution were upscaled to 0.5° x 0.5° by computing the mean across respective grid cells. The conducted multivariate analyses were restricted to time periods where values from all involved data products were available.

**Datasets**. Runoff: We employ daily stream-gauge measurements from 436 near-natural (i.e., with no or negligible human influence) catchments distributed across Europe[22]. The data range from 1984 until 2007.

Evapotranspiration: Gridded evapotranspiration computed with the Global Land Evaporation Amsterdam Model (GLEAM) is used from the version 3a GLEAM dataset[24] where evapotranspiration is computed based on reanalysis net radiation and air temperature, satellite and gauged-based precipitation, VOD, soil moisture, and snow water equivalent. The dataset is available for the time period 1980–2014, and has a spatial resolution of 0.25° x 0.25°.

For comparison, we also use evapotranspiration data from the ERA-Interim/Land reanalysis[32], which is available through 1979–2010, and has a spatial resolution of 0.5° x 0.5° as well as eddy-covariance-based measurements of ET from Fluxnet towers (http://fluxnet.fluxdata.org/data/fluxnet2015-dataset/).

Precipitation: Gridded precipitation data from the E-OBS dataset[23] is derived through interpolation and upscaling of numerous station measurement time series across Europe. The data is available through 1950–2016 and has a spatial resolution of 0.5° x 0.5°. As observed precipitation is known to underestimate the actual amount[33], we upscale all precipitation values uniformly by 10% (as done also in ref. [25]).

For comparison, we also use precipitation from the ERA-Interim reanalysis[34], which covers the time period 1979–2016 and has a spatial resolution of 0.5° x 0.5°.

Soil-moisture: Gridded soil moisture data computed with the Simple Water Balance Model (SWBM)[35] is used from the SWBM-Dataset[25]. The soil moisture computation is based on E-OBS precipitation and temperature data, as well as satellite-derived net radiation. The dataset covers Europe and is available through 1984–2013 and has a spatial resolution of 0.5° x 0.5°.

For comparison, we also use soil moisture data from the ERA-Interim/Land reanalysis[32], which is available through 1979–2010, and has a spatial resolution of 0.5° x 0.5°.

Vegetation: We use satellite-derived data on vegetation activity expressed through the Normalized Difference Vegetation Index (NDVI) from the GIMMS3g dataset[36]. It is globally available through 1981–2011 and has a spatial resolution of 0.083° x 0.083°.

## Data availability

All datasets used in the current study are publicly available from the references indicated in the previous section. All data generated and/or analyzed during this study are available from the corresponding author on reasonable request.

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

## Acknowledgements

This study was supported by funding from the Swedish Research Council Formas (grant number 2016-02045), the Swiss National Science Foundation (Advanced PostDoc. Mobility grant number 171244), and the German Research Foundation (Emmy Noether grant number 391059971). We acknowledge the E-OBS dataset from the EU-FP6 project ENSEMBLES ([http://ensembles-eu.metoffice.com], accessed 8 June 2017) and the data

providers in the ECA&D project ([http://www.ecad.eu], accessed 8 June 2017). The runoff data stem from a dataset compiled by Stahl et al. (2010), who collected data from the European water archive ([http://www.bafg.de/GRDC/], accessed 8 June 2017), from national ministries and meteorological agencies, as well as from the WATCH project ([http://www.eu-watch.org], accessed 8 June 2017).

## Author contributions

R.O. performed the analyses for the Figures. Both authors conceived the study and analyses, and contributed to the writing.

## Additional information

**Competing interests:** The authors declare no competing interests.

