## [Peer Review File · Nature Communications]

Reviewer #1 (Remarks to the Author):

Review of "Droughts are climate-driven for green while always strong and fast for blue water" by Orth and Destouni

The study by Orth and Destouni deals with the changes in water balance partitioning that occur during drought and their implications for so-called blue and green water. This is a relevant and timely topic, given the rapidly growing attention for the functional roles of green and blue water in the hydrological cycle. The authors assess these effects using a suite of what can be considered as state-of-the-art observational datasets and models. While I was initially somewhat intrigued by the idea behind the work, the manuscript fails to convince in the discussion and interpretation of the results, which I believe are at least partly flawed. The authors claim to investigate drought effect on green and blue water, which are typically defined as the amount of soil moisture available for evapotranspiration and the amount of water in groundwater and surface water reservoirs (as in a recent study by Velpuri and Senay, 2017). In their study, the authors frequently refer to the effects on blue and green water stores, whereas what they study is in fact the fluxes (ET and runoff) coming from those reservoirs. These are obviously not the same, and using them interchangeably can lead to conflicting and inconsistent statements. For instance, the title and abstract contain the message that impacts of drought on blue water (through runoff) are always strong and fast, yet the runoff anomalies during drought as shown in Fig. 3 are classified as "dampening" rather than Intensifying. This is because a large anomaly in runoff will rapidly counteract a lack to precipitation, and thus effectively "preserving" the remaining blue water store. In my view, this invalidates the main conclusions. The manuscript would need considerable rewriting to account for this shortcoming, and the conclusions of a reworked manuscript might no longer warrant publication in a high-impact journal such as Nature Communications. Especially, because some of the key points have already been highlighted in some other studies that the authors seem to have missed (Figure 3 for instance bears a strong resemblance to Figure 3 of Teuling et al., 2013, even though that was based on a much smaller subset of data). Therefore, I feel that the current manuscript is insufficiently mature to be suited for publication in Nature Communications.

References

Teuling, A. J., A. F. Van Loon, S. I. Seneviratne, I. Lehner, M. Aubinet, B. Heinesch, C. Bernhofer, T. Grünwald, H. Prasse, and U. Spank (2013), Evapotranspiration amplifies European summer drought, *Geophys. Res. Lett.*, 40, 2071–2075, doi:10.1002/grl.50495.

Velpuri, N. M. and G. B. Senay (2017), Partitioning Evapotranspiration into Green and Blue Water Sources in the Conterminous United States. *Scientific Reports* 7, 6191, doi:10.1038/s41598-017-06359-w.

Reviewer #2 (Remarks to the Author):

The paper presented by Orth and Destouni is focussing on the question whether Runoff (R) or Evapotranspiration (ET) is responding more quickly to extremely low soil-moisture conditions. For doing so, the authors evaluate a range of in-situ observations, remote-sensing based data-products and model estimates of relevant variables in Europe. The authors report that they find that R is responding more quickly at the analysed twice-monthly time-scale.

MAJOR COMMENTS:

(1) Structure and presentation:

Overall the paper has a clear scope and the individual text-blocks are written well. However, I

found that the structure of the paper made it difficult to fully grasp how the conclusions are supported by the reported quantitative evidence. For example, Figure 2 presents the conclusions of the paper (where do the numbers come from?) but figures presenting the evidence are only presented in the subsequent figures. Consequently, I have found it difficult to fully distinguish between (i) hypothesised effects, (ii) assumptions, (iii) claims and (iv) claims supported by empirical evidence within a reasonable amount of time. Therefore, I was also not able to fully grasp the strength of the arguments and to judge upon the reliability of the findings. To mitigate this issue, the paper needs to be re-structured, focussing on a clear separation of hypothesis, quantitative evidence and conclusions, alongside an unambiguous presentation of the numerical results.

(2) Novelty and links to previous research:

2.a) While I acknowledge that the format of the presented paper and the selected journal do not allow for an extensive literature survey, I am wondering why the authors have not referred to comprehensive reviews focussing on drought and drought propagation (e.g. van Loon, 2015, doi: 10.1002/wat2.1085). By doing so, they might have missed previous research on how precipitation anomalies propagate to soil-moisture, runoff and evapotranspiration. Just as an indication: van Loon (2015) mentions Evapotranspiration 23 times and runoff 22 times.

2.b) The authors make use of the blue-green water paradigm to support their claim that Evapotranspiration (ET) is (almost) exclusively related to vegetation processes, without further discussing the underlying assumptions. However, recent research has highlighted profound uncertainties in estimates of the fraction of ET that is transpired by plants (e.g. Sutanto et al, 2014, doi: 10.5194/hess-18-2815-2014), which in turn challenges the view that ET is dominated by biosphere processes.

2.c) The approach of the analysis has surprising similarities with a recent study (Nicolai-Shaw, 2017, doi:10.1016/j.rse.2017.06.014). Nicolai-Shaw (2017) use a compositing approach to analyse the response of a range of variables (including ET and vegetation indices) to extremely dry soil moisture conditions. Beside the clear technical similarity, Orth and Destouni also chose to present some of their results in a similar format (compare e.g. Fig 5 in Nicolai-Shaw (2017) with Fig 3 in the paper currently under review). Given the similarity, and the fact that Orth and Nicolai-Shaw have recently been working at the same institute (a quick google search helps to resolve this), it is hard to believe that the methods used in the presented paper were at least not inspired by the work of Nicolai-Shaw (2017).

(3) Observational data considered:

For their analysis Orth and Destouni consider a wide range of data including: (i) model estimated for soil moisture, (ii) remote sensing based information for ET and vegetation activity and (iii) station observations of river flow. As already noted by the authors, the choice of data-product can have profound impacts on their analysis. The authors do some checking (see Supporting information), but the following important questions remain:

Q1: The authors repeat their analysis using ET from three Fluxnet stations in Europe. Why was the analysis not conducted using the remaining Fluxnet stations? To the external viewer, the chosen approach seems like cherry-picking?

Q2: The analysis was conducted using soil-moisture simulations developed by the lead-author. As simulations are always prone to modelling uncertainties, I wonder why not other state-of-the-art simulations were considered. In addition, recent compilations of remote-sensing based soil moisture estimates (e.g. <http://www.esa-soilmoisture-cci.org/>) and collections of station-based soil moisture data (Dorigo et al., 2017, doi: 10.5194/hess-15-1675-2011) might have been considered.

Q3: The authors utilize station observations of streamflow to assess how unusually dry soil-moisture conditions affect runoff. In the context of the presented analysis, the choice to focus on station data for streamflow seems somewhat arbitrary given the fact that the remainder of the analysis utilizes modelled (soil moisture) and remote-sensing based (ET) products.

MINOR COMMENTS:

- 1) Clarify from early on that the focus is on soil-moisture (or agricultural) drought
- 2) lines 69-70: I think the sentence is incomplete, why is "soil-moisture ... by definition close to normal ... at the beginning". Do the authors refer to anomalies? If yes, how are they quantified?
- 3) lines 74-75: "(ii) the 10 strongest..."; I don't get it. Please reformulate more precisely.
- 4) line 77: what is "normal"? please specify.
- 5) line 87-90: Why is this surprising? Given the fact that hydrological modelling has largely focussed on rainfall-runoff processes it might just be expected that runoff is responding quickly to precipitation (and thus soil-moisture). In addition: in many hydrological models runoff is often calculated from a vertical water balance (spatial scale 10^0 m) and the kinematic-wave propagation in channels (larger spatial scale) is treated at separately and does hence not influence the total amount of water being moved.
- 6) line-numbers not available: "blocking of the water cycle"; Is this not just simply related to differences in saturated and non-saturated hydraulic conductivity which is e.g. also in practical use for "capillary barriers" in hydraulic engineering?
- 7) Figure S3: Why is net-radiation omitted from the figures in the main article? Especially given the fact that it constitutes one of the main drivers of ET.
- 8) Figure 4: What is exactly shown? Means? Medians? Sums?
- 9) Figure 3: Colours are difficult to distinguish. Sth is wrong with the y-axis (only positive numbers but zero in the centre).

Reviewer #3 (Remarks to the Author):

Summary

I like the premise for this analysis, I think it is an enlightening way of looking at droughts that can provide conclusions that are relevant for a number of applications. However, I think the title is a little misleading as this conclusion has only been shown for Europe in this study.

This study is likely to lead to a similar kind of analysis being performed for other regions, but I'd like to see a little more justification of the methodological approach, and some more discussion of the limitations of carrying out the analysis in this way.

General Comments

There is no justification as to why drought it defined as the 10 strongest soil moisture droughts in the 24 year period. This isn't really a selection of only the most extreme events. In addition, only choosing the driest half-monthly soil moisture ignores that drought impacts often related to the length of the drought rather than just the biggest anomaly. What are the assumptions and

limitations of this choice?

Most of the catchments are under 1000km² in area. Would the same conclusions apply to larger catchments?

I would like some recognition, or even further analysis, that ground-water dominated catchments may respond quite differently to runoff dominated rivers. The cited Stahl et al. (2010) paper acknowledges the buffering impact of groundwater storage in moderating climate drivers, and it may be that the conclusion summarised by Fig 2 is not valid for groundwater catchments.

Figure 2: I like this as a summary but it should be made clear within the Figure name that this summary can only be applied to this European assessment. Particularly the y-axis of climate dryness: are the stations used in this study representative of the range of dryness values globally? Presumably not. In terms of the timing of onset in the different geospheres, it might be interesting (especially for the potential applications) to consider how long it takes for them to recover as well.

Figure 3: I'm not all that fond of this because I think it overly simplifies the stations used. I think that the choice of the three European climate regions requires justification beyond just a reference to an IPCC report. It seems that the dryness index itself shows that there is significant variation within these climate regions, especially for the Centre region. It would probably be better to divide the study locations into three categories based on their dryness index (in doing this it would relate more to Figure 4 too).

Supplementary Figure 3: Runoff data are also available from the ERA-Interim Land Archive: <http://apps.ecmwf.int/datasets/data/interim-land/type=fc/>, this Figure and other supplementary Figures should be updated using these data. Note that river flow reanalyses data driven by ERA-Interim Land are also available from ECMWF.

Impact pathways across geospheres and early-warning potential: I think this as a concluding paragraph is a bit lazy; it could apply to any drought paper! The introduction mentions that 'Insufficient understanding and quantification of the drought-impact partitioning between blue and green water resources, and of its variation under different climate conditions may therefore seriously mislead or impede relevant societal responses to drought and its possible future intensification.' So do the conclusions from this paper now change things? How should they be used?

With respect to drought early warning, meteorological / hydrometeorological forecasting systems are also required on top of drought monitoring to provide the early warning required.

We are thankful to all reviewers for their valuable feedback which helped us to improve the manuscript. In response, aside from several minor corrections, we have introduced the following main changes to the paper:

(1) We have clarified in all parts of the manuscript, and particularly in the abstract, that (i) we are focusing on Europe while similar analyses for other regions are yet to be done, and that (ii) we are mainly investigating blue and green water *fluxes*, including associated implications for net total *change in water storage* (which is also a flux, volume change per time), rather than blue and green water storages (volumes).

(2) We have re-structured the manuscript to improve readability and separation between actual results and interpretation of them. The conceptual summary of our results formerly provided at the beginning and in Figure 2 (new Figure 5) has been moved to the end of the manuscript, just before the conclusions.

(3) Motivated by the reviewers' ideas we have added several interesting side analyses. For example, we have re-computed the drought temporal evolution figure (Figure 2, previously Figure 3) for catchments grouped with respect to dryness index (new Figure S1), as well as separately for small and large catchments (new Figure S2).

(4) We have added several references to relevant literature as suggested by the reviewers to better link this study to other state-of-the-art literature.

Reviewer #1

The study by Orth and Destouni deals with the changes in water balance partitioning that occur during drought and their implications for so-called blue and green water. This is a relevant and timely topic, given the rapidly growing attention for the functional roles of green and blue water in the hydrological cycle. The authors assess these effects using a suite of what can be considered as state-of-the-art observational datasets and models.

A1: We thank the reviewer for underlining the importance of the focus topic of our study.

While I was initially somewhat intrigued by the idea behind the work, the manuscript fails to convince in the discussion and interpretation of the results, which I believe are at least partly flawed. The authors claim to investigate drought effect on green and blue water, which are typically defined as the amount of soil moisture available for evapotranspiration and the amount of water in groundwater and surface water reservoirs (as in a recent study by Velpuri and Senay, 2017). In their study, the authors frequently refer to the effects on blue and green water stores, whereas what they study is in fact the fluxes (ET and runoff) coming from those reservoirs. These are obviously not the same, and using them interchangeably can lead to conflicting and inconsistent statements. For instance, the title and abstract contain the message that impacts of drought on blue water (through runoff) are always strong and fast, yet the runoff anomalies during drought as shown in Fig. 3 are classified as “dampening” rather than Intensifying. This is because a large anomaly in runoff will rapidly counteract a lack to precipitation, and thus effectively “preserving” the remaining blue water store.

A2: We thank the reviewer for emphasizing the need to better clarify water fluxes and storages. We investigate here blue and green water fluxes rather than blue and green water storages. This focus is physically appropriate and relevant, as water flux (volume per time) rather than water storage (volume) determines the renewable water resource, which is also why blue and green water availability is defined in terms of the fluxes of blue and green water that the (undifferentiated) precipitation flux partitions into (e.g., in reference [4] of the revised manuscript and in the glossary of the international Water Footprint Network - <http://waterfootprint.org/en/water-footprint/glossary/>).

Furthermore, a change in storage is also a flux (volume change per time in soil moisture, groundwater, surface water storage), so referring interchangeably to fluxes and storage changes is indeed physically appropriate and necessary, as both are components in the fundamental water balance of a catchment. Regarding the mentioned Fig. 3 (now Fig. 2 in the revised manuscript), even though soil moisture is here illustrated in storage terms (standard deviation during drought relative to normal, long-term average soil moisture), this is done versus drought evolution time so that that the change per time (a flux) can also be inferred from the illustrated soil moisture results. With regard to the term “dampening” in the context of this figure, this refers to the drought-induced runoff decrease counteracting drought-induced storage depletion, as a direct consequence of fundamental catchment water balance. In the latter, associated precipitation, evapotranspiration, runoff and storage-change responses during drought evolution are linked and must be compared, and this does not in any way lead to conflicting and inconsistent statements.

We have clarified these points throughout the manuscript, for example:

- title:

“Drought impacts are climate-driven for green while always strong for blue water fluxes”

- abstract (lines 17-21):

“... resolving the partitioning of water deficit during drought into blue-water runoff and green-water evapotranspiration fluxes is critical, as anomalies in these fluxes threaten different associated societal sectors and ecosystems. We here analyze the propagation of drought-inducing precipitation deficits through soil moisture reductions to their impacts on blue and green water fluxes ...”

- introduction (lines 41-42 and lines 45-46):

“However, the partitioning of drought-related water deficits between green (ET) and blue water (runoff) water fluxes remains largely unresolved ...”

“Specifically, green water flux anomalies are primarily associated with vegetation impacts ...”

- main text (lines 101-106):

“Note that, by fundamental water balance, any net imbalance of input (precipitation) and output (ET, runoff) fluxes occurring at any point in time during the drought period must correspond to an associated change in water storage within the catchment at that time. The generally strong and persistent drought reduction of blue water flux (runoff) counteracts then water storage depletion, while the relatively unaffected green water flux (ET) promotes it.”

- conclusions (lines 234-235):

“... and revealed essential early-warning and mitigation-adaptation potential, in that blue and green water fluxes are impacted weeks or months after drought initiation, respectively.”

In my view, this invalidates the main conclusions. The manuscript would need considerable rewriting to account for this shortcoming, and the conclusions of a reworked manuscript might no longer warrant publication in a high-impact journal such as Nature Communications.

A3: As also explained in our response A2 above, the reviewer expresses here misunderstanding of the meaning and our consideration of “storage” and “storage change”. We consider the latter, which is in fact a flux and, by fundamental water balance, directly linked and comparable with the other investigated water fluxes in this study. We strongly disagree with the reviewer regarding the conclusions, which are indeed consistent and valid for both blue and green water fluxes, and their implications for water storage change. The fluxes determine the renewable green and blue water resources, which is also why they are used to define green and blue water availability, e.g., by the international Water Footprint Network (<http://waterfootprint.org/en/water-footprint/glossary/>). Furthermore, the reviewer’s difficulty to realize the direct linkages of and the associated drought impact partitioning among green and blue water fluxes and storage changes only emphasizes the novelty and significance of our study and results. The results are essential for understanding and distinguishing drought impacts on, e.g., crop yields and near-surface temperature/heat exchange and balance (related to green-water flux anomalies), and irrigation, energy, industrial and household water uses (related to blue-water flux anomalies). Hence, we remain convinced that revealing the drought-impact pathways through blue and green water fluxes, and associated water storage changes, as done in our study, is a key novel contribution to inform and guide society about drought evolution and its impacts.

Especially, because some of the key points have already been highlighted in some other studies that the authors seem to have missed (Figure 3 for instance bears a strong resemblance to Figure 3 of Teuling et al., 2013, even though that was based on a much smaller subset of data). Therefore, I feel that the current manuscript is insufficiently mature to be suited for publication in Nature Communications.

A4: We thank the reviewer for pointing us to this reference, which indeed provides support for our results. We have added it to the manuscript in lines 206-209. Note, however, that the analysis of Teuling et al. is entirely site- and time-specific and far less comprehensive and general than our study: They only consider hydrological data over one year at a particular site, whereas we consider decadal, continental-scale datasets, which allow us to infer the hydrological and biospheric drought response across different climate regimes. Moreover, we also provide quantification of typical drought-related water anomalies in an absolute sense, which can serve as a direct basis for adaptation and mitigation strategies.

Lines 206-209:

“Similar results to Figure 4 are also obtained with (i) alternative gridded data on precipitation, ET and soil moisture (Figure S7), and these general results are also consistent with (ii) site- and time-specific results of an earlier study investigating the 2003 summer at the Swiss Rietholzbach site (31). ”

References

Teuling, A. J., A. F. Van Loon, S. I. Seneviratne, I. Lehner, M. Aubinet, B. Heinesch, C. Bernhofer, T. Grünwald, H. Prasse, and U. Spank (2013), Evapotranspiration amplifies European summer drought, *Geophys. Res. Lett.*, 40, 2071–2075, doi:10.1002/grl.50495.

Velpuri, N. M. and G. B. Senay (2017), Partitioning Evapotranspiration into Green and Blue Water Sources in the Conterminous United States. *Scientific Reports* 7, 6191, doi:10.1038/s41598-017-06359-w.

Reviewer #2

The paper presented by Orth and Destouni is focussing on the question whether Runoff (R) or Evapotranspiration (ET) is responding more quickly to extremely low soil-moisture conditions. For doing so, the authors evaluate a range of in-situ observations, remote-sensing based data-products and model estimates of relevant variables in Europe. The authors report that they find that R is responding more quickly at the analysed twice-monthly time-scale.

MAJOR COMMENTS:

(1) Structure and presentation:

Overall the paper has a clear scope and the individual text-blocks are written well.

B1: We thank the reviewer for this encouraging comment.

However, I found that the structure of the paper made it difficult to fully grasp how the conclusions are supported by the reported quantitative evidence. For example, Figure 2 presents the conclusions of the paper (where do the numbers come from?) but figures presenting the evidence are only presented in the subsequent figures. Consequently, I have found it difficult to fully distinguish between (i) hypothesised effects, (ii) assumptions, (iii) claims and (iv) claims supported by empirical evidence within a reasonable amount of time. Therefore, I was also not able to fully grasp the strength of the arguments and to judge upon the reliability of the findings. To mitigate this issue, the paper needs to be re-structured, focussing on a clear separation of hypothesis, quantitative evidence and conclusions, alongside an unambiguous presentation of the numerical results.

B2: Following the reviewer's suggestion, we have re-structured the manuscript. Figure 5 (previously Figure 2) and the corresponding discussion are moved to the end of the manuscript, and serve now as a summary before the conclusions are presented. With this updated structure, we believe that the empirical evidence presented in Figures 1-4 is sufficiently separated from the mechanistic interpretation and summary shown thereafter in the new Figure 5.

The numbers mentioned by the reviewer have been removed from Figure 5 (previously Figure 2).

(2) Novelty and links to previous research:

2.a) While I acknowledge that the format of the presented paper and the selected journal do not allow for an extensive literature survey, I am wondering why the authors have not referred to comprehensive reviews focussing on drought and drought propagation (e.g. van Loon, 2015, doi: 10.1002/wat2.1085). By doing so, they might have missed previous research on how precipitation anomalies propagate to soil-moisture, runoff and evapotranspiration. Just as an indication: van Loon (2015) mentions Evapotranspiration 23 times and runoff 22 times.

B3: We thank the reviewer for pointing us to this reference. We have included it in the very beginning of the manuscript (lines 32), and also later in the introduction (line 43). The reference fits particularly well in the latter position as it nicely highlights current limitations in the understanding of drought propagation across geospheres and calls for more comprehensive analyses to assess and compare the role of evapotranspiration – just as we do in our study.

2.b) The authors make use of the blue-green water paradigm to support their claim that Evapotranspiration (ET) is (almost) exclusively related to vegetation processes, without further discussing the underlying assumptions. However, recent research has highlighted profound uncertainties in estimates of the fraction of ET that is transpired by plants (e.g. Sutanto et al, 2014, doi: 10.5194/hess-18-2815-2014), which in turn challenges the view that ET is dominated by biosphere processes.

B4: Our results in Figure 3 tend to underline an important role of transpiration for total European summer ET. However, addressing the reviewers comment we acknowledge that the contribution of transpiration to ET is variable across regions and seasons, and that the exact overall contribution is still under debate.

Lines 177-180:

“The ET responses to droughts of different strength are further largely consistent with those of vegetation activity (Figure 3c). This underlines an important role of plant transpiration for ET across Europe (29), even though the contribution of transpiration to ET varies across time and space, and is still under debate (30).”

2.c) The approach of the analysis has surprising similarities with a recent study (Nicolai-Shaw, 2017, doi:10.1016/j.rse.2017.06.014). Nicolai-Shaw (2017) use a compositing approach to analyse the response of a range of variables (including ET and vegetation indices) to extremely dry soil moisture conditions. Beside the clear technical similarity, Orth and Destouni also chose to present some of their results in a similar format (compare e.g. Fig 5 in Nicolai-Shaw (2017) with Fig 3 in the paper currently under review). Given the similarity, and the fact that Orth and Nicolai-Shaw have recently been working at the same institute (a quick google search helps to resolve this), it is hard to believe that the methods used in the presented paper were at least not inspired by the work of Nicolai-Shaw (2017).

B5: We have updated the description of the drought composite computation to include a reference to the similar approach taken by Nicolai-Shaw et al. (2017) in lines 83-86:

“Using precipitation, runoff and ET data in these mean drought periods, we compute composites across all considered droughts to study temporal drought evolution (as also done in 27).”

(3) Observational data considered:

For their analysis Orth and Destouni consider a wide range of data including: (i) model estimated for soil moisture, (ii) remote sensing based information for ET and vegetation activity and (iii) station observations of river flow. As already noted by the authors, the choice of data-product can have profound impacts on their analysis. The authors do some checking (see Supporting information), but the following important questions remain:

Q1: The authors repeat their analysis using ET from three Fluxnet stations in Europe. Why was the analysis not conducted using the remaining Fluxnet stations? To the external viewer, the chosen approach seems like cherry-picking?

B6: We thank the reviewer for this comment. We clarify this point in the caption of the corresponding Figure S3 (previously Figure S2) in lines 589-590:

“Fluxnet sites with most available ET data were chosen in each region to ensure a robust computation of the drought composite ET.”

Q2: The analysis was conducted using soil-moisture simulations developed by the lead-author. As simulations are always prone to modelling uncertainties, I wonder why not other state-of-the-art simulations were considered. In addition, recent compilations of remote-sensing based soil moisture estimates (e.g. <http://www.esa-soilmoisture-cci.org/>) and collections of station-based soil moisture data (Dorigo et al., 2017, doi: 10.5194/hess-15-1675-2011) might have been considered.

B7: We agree with the reviewer that modeled datasets are inherently uncertain as a model is always a simplification of nature. Addressing this point we have re-computed our analysis with independent soil moisture and ET data from the ERA-Interim/Land dataset. The result is displayed in Figure S7 (previously Figure S6), and the obvious similarity with the results in Figure 4 (previously Figure 5) highlights the robustness of our results. To clarify this point we have rephrased the corresponding paragraph in the manuscript in lines 206-210:

“Similar results to Figure 4 are also obtained with (i) alternative gridded data on precipitation, ET and soil moisture (Figure S7), and these general results are also consistent with (ii) site- and time-specific results of an earlier study investigating the 2003 summer at the Swiss Rietholzbach site (31). This highlights the robustness of our findings as they are valid across independent datasets.”

Moreover, as we infer droughts from the soil moisture data in this study, we require long-term and (mostly) gap-free soil moisture records. Hence, we can unfortunately not consider measured station-based soil moisture data, and have to employ modeled products instead.

Further, remote sensing-based soil moisture is also not suitable for our analyses. It only captures the water contained in the uppermost centimeters of the soil. For our analyses this is insufficient as we are considering root-zone soil water content (contributing to ET) as well as the water content in the saturated zone (contributing to runoff). We therefore require total-column soil moisture estimates.

Q3: The authors utilize station observations of streamflow to assess how unusually dry soil-moisture conditions affect runoff. In the context of the presented analysis, the choice to focus on station data for streamflow seems somewhat arbitrary given the fact that the remainder of the analysis utilizes modelled (soil moisture) and remote-sensing based (ET) products.

B8: We think that station data in general comes closest to ground truth. Therefore, we try to employ station data where-ever possible. For this study, this was only possible in the case of runoff as (i) the corresponding measurements cover a long time-span (24 years) and (ii) there is a sufficient number of stations (>400) covering most of the European continent. As these conditions are not fulfilled for station data of soil moisture (international soil moisture database, see response B7), and of ET (fluxnet towers), we instead use state-of-the-art modelling products to represent these variables.

MINOR COMMENTS:

1) Clarify from early on that the focus is on soil-moisture (or agricultural) drought

B9: We have clarified this point in lines 55-57:

“In this study, we focus on soil-moisture drought periods. Therein, we analyze the propagation of precipitation deficits through soil moisture decreases to associated changes in runoff and ET ...”

2) lines 69-70: I think the sentence is incomplete, why is “soil-moisture ... by definition close to normal ... at the beginning”. Do the authors refer to anomalies? If yes, how are they quantified?

3) lines 74-75: “(ii) the 10 strongest...”; I don’t get it. Please reformulate more precisely.

4) line 77: what is “normal”? please specify.

B10: We have rephrased and re-structured this paragraph to address the reviewer's comments in lines 70-83:

“We quantify drought in terms of soil moisture anomalies, a simple and relevant drought index (1). Anomalies in this study are computed by removing the mean seasonal cycle from the actual time series at each catchment. We analyze composites of the 10 strongest soil-moisture droughts (greatest reductions from seasonal mean soil moisture at each time) occurring in each catchment during the 24-year period 1984-2007. While there are numerous ways to quantify drought strength (1,2) we focus on the size of the maximum soil moisture anomaly. For this purpose, (i) the driest half-monthly, total-column soil-moisture anomaly in the warm season (May-September) of each year is identified, yielding 24 annual driest anomalies. From these, (ii) the 10 strongest anomalies are selected, and finally (iii) the mean drought period is determined as the total time before drought peak (drought buildup period) and after drought peak (recovery period), during which soil moisture is reduced below normal in the majority of the selected 10 drought years. Soil moisture levels are therefore close to normal (i.e., unchanged by drought and thus zero anomaly) at the beginning and end of each drought period, such that changes in soil-water storage over the period can be neglected.”

5) line 87-90: Why is this surprising? Given the fact that hydrological modelling has largely focussed on rainfall-runoff processes it might just be expected that runoff is responding quickly to precipitation (and thus soil-moisture). In addition: in many hydrological models runoff is often calculated from a vertical water balance (spatial scale 10^0 m) and the kinematic-wave propagation in channels (larger spatial scale) is treated separately and does hence not influence the total amount of water being moved.

B11: We have removed the statement that this finding is surprising in line 224-225:

“This key finding represents a spatio-temporal paradox ...”

6) line-numbers not available: “blocking of the water cycle”; Is this not just simply related to differences in saturated and non-saturated hydraulic conductivity which is e.g. also in practical use for “capillary barriers” in hydraulic engineering?

B12: We thank the reviewer for raising this point, and have included it in the manuscript in lines 107-112:

“This illustrates a blocking of the water cycle, which the excess precipitation water in this drought phase largely contributing to refill the subsurface water storage, without substantially adding to runoff. As soil moisture increases and the groundwater table is raised in this process, also groundwater runoff must increase due to higher saturated than unsaturated hydraulic conductivity of soils combined with increased hydraulic gradient towards surface water.”

7) Figure S3: Why is net-radiation omitted from the figures in the main article? Especially given the fact that it constitutes one of the main drivers of ET.

B13: While net radiation is mentioned in the main text (for example in lines 132 and 181), it is not part of the main figures. This is to keep the main figures as simple and straightforward as possible while including important supporting information in the supplementary material.

8) Figure 4: What is exactly shown? Means? Medians? Sums?

B14: We have clarified this point in the caption of Figure 3 (previously Figure 4) in line 404-406:

“Values shown in each box are means across all droughts characterised by the respective precipitation deficit and occurring in catchments with respective long-term dryness.”

9) Figure 3: Colours are difficult to distinguish. Sth is wrong with the y-axis (only positive numbers but zero in the centre).

B15: We checked on different computer systems/screens and feel that the colors can be sufficiently distinguished. Signs at the y-axis are omitted on purpose as any shown anomaly is classified into dampening or intensifying drought. Leaving out signs in this context helps to avoid confusion, we hope, as the signs would be opposite for e.g. precipitation and runoff anomalies.

Reviewer #3

Summary

I like the premise for this analysis, I think it is an enlightening way of looking at droughts that can provide conclusions that are relevant for a number of applications.

C1: Many thanks for this encouraging comment.

However, I think the title is a little misleading as this conclusion has only been shown for Europe in this study.

C2: We agree with the reviewer that this is an important information. However, due to length restrictions we cannot expand the title to include it. Instead we have emphasized the geographical focus of the study in the abstract:

lines 19-22:

“We here analyze the propagation of drought-inducing precipitation deficits through soil moisture reductions to their impacts on blue and green water fluxes ... across Europe.”

lines 22-24:

“We show that the soil-moisture drought propagation reduces runoff stronger and faster than ET over the entire continent.”

lines 25-27:

“ Understanding these drought-impact pathways on blue and green water fluxes and across geospheres in Europe and beyond is essential for our ability to ensure food and water security ...”

Moreover, we have once again included this information in the conclusions section in lines 251-253:

“While the study focus has been on Europe and other regions remain yet to be investigated, the present findings should trigger more exchanges and joint efforts of different geosphere science communities.”

This study is likely to lead to a similar kind of analysis being performed for other regions, but I'd like to see a little more justification of the methodological approach, and some more discussion of the limitations of carrying out the analysis in this way.

C3: See responses C4-C6 below.

General Comments

There is no justification as to why drought is defined as the 10 strongest soil moisture droughts in the 24 year period. This isn't really a selection of only the most extreme events. In addition, only choosing the driest half-monthly soil moisture ignores that drought impacts often related to the length of the drought rather than just the biggest anomaly. What are the assumptions and limitations of this choice?

C4: We agree with the reviewer that the selection of 10 droughts is somewhat arbitrary. Figure 3 (previously Figure 4) therefore also presents results for droughts of different magnitude, and shows that our main conclusion of runoff being more strongly affected than ET is valid across all droughts. Hence, the number and related severity of the chosen droughts is not impacting our conclusions. We have added this argument to the manuscript in lines 172-176:

"... Figure 3 shows that blue water runoff is generally more strongly affected by drought than green water ET in all considered climate regimes and for droughts of all investigated magnitudes. The latter further indicates that the (necessarily) arbitrary selection of the 10 strongest droughts in the above drought-evolution part of this study should not greatly impact the related results and conclusions."

Furthermore, we clarify that the driest soil moisture anomaly is used to characterize drought strength in this study in lines 74-75:

"While there are numerous ways to quantify drought strength (1,2) we focus on the size of the maximum soil moisture anomaly."

In addition, the results in Figure 3 (previously Figure 4) suggest that different measures of drought strength are related such that any choice between them is expected to yield similar conclusions (lines 193-197):

"Moreover, drought duration is related to drought strength expressed through the associated precipitation deficit (Figure 3d). Such a relationship across different measures of drought strength suggests that our conclusions are insensitive to choices of such strength measures for characterizing droughts."

Most of the catchments are under 1000km² in area. Would the same conclusions apply to larger catchments?

C5: We added a corresponding analysis to the manuscript with the new Figure S2, and find similar results across small and large catchments as discussed in lines 115-118:

"Also, similar results to Figure 2 are obtained across catchments of different size as shown in Figure S2 for the smallest (size < 50 km²) and largest (size > 1000 km²) considered catchments, even though the runoff response is slightly faster for the small catchments."

I would like some recognition, or even further analysis, that ground-water dominated catchments may respond quite differently to runoff dominated rivers. The cited Stahl et al. (2010) paper acknowledges the buffering impact of groundwater storage in moderating climate drivers, and it may be that the conclusion summarised by Fig 2 is not valid for groundwater catchments.

C6: We thank the reviewer for this comment, and have added discussion on this point to the manuscript. We agree that aquifer characteristics and corresponding groundwater availability are important for the catchment's drought response. This can probably explain the considerable spatial variability in runoff

response to drought displayed in Figure S6, which is not entirely explained by climate characteristics, in contrast to the ET response. Consequently, we state the conclusions of our analysis hold in most regions across the continent but we cannot exclude that some special local aquifer characteristics can lead to different behavior.

Lines 166-174:

“While most of the ET anomaly variability is explained by different dryness indices and precipitation deficits (Figure S6d), this is not the case for runoff (Figure S6b). The remaining runoff variability may depend on different aquifer characteristics, yielding different groundwater storage and flow changes and thereby different runoff responses to drought in different catchments. Even though such aquifer variability can lead to locally different drought responses in blue vs. green water fluxes, Figure 3 shows that blue water runoff is generally more strongly affected by drought than green water ET in all considered climate regimes and for droughts of all investigated magnitudes.”

Figure 2: I like this as a summary but it should be made clear within the Figure name that this summary can only be applied to this European assessment. Particularly the y-axis of climate dryness: are the stations used in this study representative of the range of dryness values globally? Presumably not. In terms of the timing of onset in the different geospheres, it might be interesting (especially for the potential applications) to consider how long it takes for them to recover as well.

C7: We have added this point to the caption of Figure 5 (previously Figure 2) in lines 423-424:

“This summary is valid across the investigated European climate regimes and drought strengths.”

Further, we thank the reviewer for the comment on the recovery times and have added corresponding discussion to the manuscript in lines 233-238:

“Moreover, the longer a precipitation deficit persists and the further drought consequently propagates across geospheres, the longer it takes for the hydrosphere water to recover to its normal magnitude (in soil moisture and runoff, Figures 2 and 3). In contrast, the ET and associated plant transpiration of the biosphere recover almost immediately after drought peak. It remains to be further investigated if the latter behavior would also occur in drier climate regimes outside Europe.”

Figure 3: I’m not all that fond of this because I think it overly simplifies the stations used. I think that the choice of the three European climate regions requires justification beyond just a reference to an IPCC report. It seems that the dryness index itself shows that there is significant variation within these climate regions, especially for the Centre region. It would probably be better to divide the study locations into three categories based on their dryness index (in doing this it would relate more to Figure 4 too).

C8: As motivated by the reviewer we have re-computed Figure 2 (previously Figure 3) for the catchments grouped with respect to dryness index, and included it as Figure S1. The results are similar as for the geographical assessment, as discussed in lines 112-115:

“Similar results as displayed in Figure 2 are also obtained when grouping the catchments with respect to dryness index (Figure S1). This highlights a dominant role of the different climates across the investigated European sub-regions in shaping the temporal drought response.”

We have decided to keep the geographical assessment in Figure 2 (previously Figure 3) alongside the dryness index-based analysis in Figure 3 (previously Figure 4) to cover both perspectives in the main

results figures of this study.

Supplementary Figure 3: Runoff data are also available from the ERA-Interim Land Archive: <http://apps.ecmwf.int/datasets/data/interim-land/type=fc/>, this Figure and other supplementary Figures should be updated using these data. Note that river flow reanalyses data driven by ERA-Interim Land are also available from ECMWF.

C9: Following the reviewer's suggestion we have updated Figure S4 (previously Figure S3) to include gridded runoff from the ERA-Interim/Land dataset. We find that the temporal drought evolution of this runoff product is comparable to the dynamics found with the stream gauge measurements (lines 122-125): "Finally, to test the spatial representativeness of our catchment sampling (which is unavoidable in order to include observed runoff in the analysis) Figure 2 is re-computed with gridded data for the whole European regions (Figure S4). Similar results are obtained, indicating robustness of the catchment-based findings."

Impact pathways across geospheres and early-warning potential: I think this as a concluding paragraph is a bit lazy; it could apply to any drought paper! The introduction mentions that 'Insufficient understanding and quantification of the drought-impact partitioning between blue and green water resources, and of its variation under different climate conditions may therefore seriously mislead or impede relevant societal responses to drought and its possible future intensification.' So do the conclusions from this paper now change things? How should they be used?

C10: We have expanded the concluding paragraph to better highlight the implications of the reported findings in lines 241-248:

"The presented results suggest that drought response measures need to be tailored to (i) climate regime, and (ii) elapsed drought duration. In particular, in wetter climate and/or early into a drought, response measures should focus on adapting to lower runoff levels, e.g. by adjusting dam operations for increased support of downstream water uses, navigation and aquatic ecosystems. In drier climate, and/or further into a drought, the focus should be extended or even shifted to targeted irrigation support of essential crops and vegetation, while balancing and temporarily limiting other water uses, and also preparing communities for higher temperatures induced by lower ET and consequently increased sensible heat flux. "

With respect to drought early warning, meteorological / hydrometeorological forecasting systems are also required on top of drought monitoring to provide the early warning required.

C11: We thank the reviewer for this comment, and have added the argument to the manuscript in lines 236-238:

"These response times can be exploited in combination with hydro-meteorological forecasting and operational drought monitoring (32) ..."

Reviewer #1 (Remarks to the Author):

Review of "Drought impacts are climate-driven for green while always strong for blue water fluxes" by Orth and Destouni

The manuscript by Orth and Destouni has improved considerably in comparison to the previous version. It now provides a more complete overview of drought impacts on water balance partitioning across different climate zones, and it includes an improved in-depth discussion. The results are original and conceptualize many of the previous work carried out at smaller scales. Most of my comments have been adequately addressed. My main remaining concern, also in line with one of the comments by referee #3, deals with the motivation for using the climate regions. While I understand that the authors want to "collapse" the geographical space for simplicity, I am not convinced with the use of only the climate regions because they contain too much spread in dryness-index values. This issue can, and in my opinion should, be solved by combining Fig 2 with Fig S1. In this way, readers can directly compare the latitudinal and climatic impacts on the dynamics (key to a good understanding of the results), without having to compare two documents.

In addition to this comments, there are a few specific comments that will need to be addressed:

Title: The title can be sharpened. Europe is not mentioned, and the current title is rather lengthy. What about something along the lines of "Contrasting green and blue water flux response to drought in Europe"?

Line 83 "so that changes in soil-water storage over the period can be neglected" -> this is incorrect. The fact that soil moisture anomalies are small at the beginning and end of drought events does not say anything about actual soil moisture.

Line 101-104 -> Yes, but this is only true if all fluxes are accounted for (including deeper flow over catchment boundaries), and of course when using independently observations of all balance components the balance will generally not close because of measurement uncertainties. Note that only precipitation is measured as a mass-flux in rain gauges; discharge is measured per volume, and evaporation often as energy flux.

Line 120 "(state of the art)" -> One could also argue that using a "bucket-style" model forced by Priestley-Taylor potential ET is not really state-of-the-art, but I agree it is one of the more reliable products available.

Reviewer #2 (Remarks to the Author):

The authors have substantially revised the manuscript and have made it thereby significantly easier to grasp the details and the extent of their analysis. While this clearly helps to understand how the conclusions are supported by the results, it also reveals weak-points of their assessment which question originality of the work as well as the robustness of their conclusions.

Issue 1:

Originality of the analysis

Large aspects of the analysis are equivalent to the analysis conducted by Nicolai-Shaw et al. (2017, doi: 10.1016/j.rse.2017.06.014), which also produce a set of figures similar to Figure 2 (and versions thereof). In particular, Figure S2 (supporting information) of Nicolai-Shaw et al. (2017) features drought response composites for P, E and NDVI for the three European macro regions. This figure also shows several features highlighted by Orth & Destouni, including e.g. the

delayed response of ET and NDVI as well as the positive NDVI anomalies prior to the strongest soil moisture anomaly.

This leaves the impression that Orth & Destouni are trying to hide issues with the originality of their analysis.

I Acknowledge that the analysis of runoff and the interpretation related to that is substantially novel.

Issue 2:

Incomplete use of considered data sets suggests either a sloppy approach or that the authors are hiding unwanted results.

This severely questions the robustness of their conclusions.

Analysing water fluxes at continental scales from an observational perspective will always be a compromise between using sparse station observations and model-based estimates which provide full spatial and temporal coverage. I, therefore, acknowledge the authors efforts to base their assessment on both station observations and model-based estimates. However, the authors are incomplete with their assessment, raising questions regarding the reliability of the conclusions. Specifically,

(2.a) Using only 3 of dozens of FLUXNET stations raises the question what happens if other stations are considered. I suspect that some of the results might vanish, questioning the conclusions drawn by the authors.

(2.b) Partial use of model estimates. The authors use model estimates of soil moisture and evapotranspiration. But why is modelled runoff not considered to augment the analysis of the runoff observations. To me this is inconsequential – if evapotranspiration and soil moisture are considered reliable enough to draw strong conclusions, the analysis should be extended to runoff simulations of all models. Again, this raises the question whether the conclusions remain robust if the full data set is considered.

(2.c) Very small and biased model ensemble. The main analysis is based on three model products (SWBM (soil-moisture), ERA-Interim/Land (soil-moisture, ET), GLEAM (ET)), which are used for different purposes of the analysis. Generally, ERA-Interim/Land is used to demonstrate the robustness of the results derived from the other products. However, considering the availability of comprehensive global hydrological model ensembles (see e.g. <http://www.earth2observe.eu/>, or ISIMIP2a doi:10.5880/PIK.2017.010) this choice appears to be limited and biased towards the authors own work (SWBM is a product developed by R. Orth). Given the large uncertainties involved in large scale hydrological modelling this raises questions w.r.t the robustness of the presented assessment which can only be resolved if a larger model ensemble is considered.

Issue 3

Over-interpretation of some of the results (examples, to illustrate the tendency

While reading the paper, I repeatedly have the impression that the authors tend to over-interpret their results, which sometimes results in erroneous conclusions

Two examples:

(3.a) The authors discuss what they refer to as “blocking of the water cycle”, and describe one plausible mechanism. However, what they observe is essentially a non-linear response of runoff to storage which in turn is common knowledge in catchment sciences (see e.g.: Kirchner (2009, doi: 10.1029/2008WR006912) or Clark et al (2011, doi: 10.1002/hyp.7902).

(3.b) In Figure 3b, they state “drought promotes biosphere”, which is plainly wrong. What the figure shows is the fact that in humid climates good weather periods foster vegetation growths. Water limitations only play a role for extremely low absolute soil moisture values. The fact that the authors hint at this mechanism in the text is acknowledged but overshadowed by the misleading

statement in the figure.

Issue 4

Some of the graphics are non-transparent, misleading or wrong

Examples

(4.a) Figure 2 remains to have a non-logical x-axis which includes only positive values but goes through zero. This leads to strange plotting behaviour with a variable starting "positive" crossing zero and going again to "positive" but on the other side of the plot. Even after repeated thinking and the authors explanation in the rebuttal I could not get my head around that. The fact that this non-physical axis is even not mentioned in the caption does not make this better, and will contribute to confusing any potential reader of this assessment.

(4.a) Figure 4 why are all values positive, even when the values in Fig 2 do cross the zero line? To me this remains un-accessible and I doubt that it will be accessible to most of the readers.

Reviewer #3 (Remarks to the Author):

I am happy that the authors have addressed my comments from the first review, though I hold firm that the title needs changing.

I think it is important that the title reflects what can be concluded from the paper. I understand the authors' comment about the word limit, but I believe that the maximum word limit for the title is 15 words, therefore the following would be fine:

"European drought impacts are climate-driven for green while always strong for blue water fluxes"

A Europe-wide analysis is still a significant achievement that is worthy for publication in Nature Communications, there is no need to overlay the results.

We appreciate the repeated feedback of all reviewers which helped us to further improve the manuscript. In response, aside from several minor corrections, we have introduced the following main changes to the paper:

(1) As requested by reviewer #1, we have merged Figures 2 and S1 into a new comprehensive Figure 2. This way, the reader can better understand that differences in hydrological and biospheric drought responses across Europe are driven by differences in climate.

(2) As requested by reviewers #1 and #3, we have revised the title to include the European focus of our study: “Contrasting green and blue water flux response to drought in Europe”

(3) As requested by reviewer #2 we have considerably increased the number of considered Fluxnet stations for the validation of the employed model-based GLEAM dataset in Figure S2. The results remain the same, even though they are now more robust thanks to more stations – the GLEAM dataset agrees sufficiently well with ground truth data and is hence suitable to be employed in our study.

(4) Also as requested by reviewer #2 we have clarified similarities and differences between our study and that of Nicolai-Shaw et al. (2017). While we use the drought composite analysis approach introduced in that study, we move beyond determining the drought evolution of particular quantities and analyse the respective impact-relevant drought-integrated anomalies.

More importantly, as also acknowledged by the reviewer, we evaluate runoff data in addition to biospheric data. Consequently, our main finding that drought has stronger and faster impacts on blue-water runoff than on green-water ET in Europe (as reflected in title, abstract, and conclusions) is not reported/mentioned/shown in any way in Nicolai-Shaw et al. (2017). This novel finding also shows that, by ignoring runoff, they actually omitted the most important part of the hydrological drought-impact partitioning and evolution.

Reviewer #1

The manuscript by Orth and Destouni has improved considerably in comparison to the previous version. It now provides a more complete overview of drought impacts on water balance partitioning across different climate zones, and it includes an improved in-depth discussion. The results are original and conceptualize many of the previous work carried out at smaller scales. Most of my comments have been adequately addressed.

A1: We thank the reviewer for the encouraging evaluation of the revised manuscript.

My main remaining concern, also in line with one of the comments by referee #3, deals with the motivation for using the climate regions. While I understand that the authors want to “collapse” the geographical space for simplicity, I am not convinced with the use of only the climate regions because they contain too much spread in dryness-index values. This issue can, and in my opinion should, be solved by combining Fig 2 with Fig S1. In this way, readers can directly compare the latitudinal and climatic impacts on the dynamics (key to a good understanding of the results), without having to compare two documents.

A2: We have implemented the reviewer's suggestion and merged Figures 2 and S1 into a new comprehensive Figure 2. We have adapted the manuscript accordingly in lines 103-109:

"Figure 2 shows the mean temporal evolution of precipitation, runoff and ET anomalies during drought, with the anomalies grouped in terms of whether they intensify or dampen drought, instead of their actual sign. Results are shown as averages across all catchments in each of the three European regions (Figure 2a-c), and as averages across all catchments located in comparable climate (i.e. dryness) conditions (Figure 2d-f). The similarity of the results in Figure 2a-c and 2d-f indicates a dominant role of the dryness of different climates across the investigated European sub-regions in shaping the temporal drought response."

In addition to this comments, there are a few specific comments that will need to be addressed:

Title: The title can be sharpened. Europe is not mentioned, and the current title is rather lengthy. What about something along the lines of "Contrasting green and blue water flux response to drought in Europe"?

A3: We have shortened the title, and included the European focus:

"Drought reduces blue much stronger than green water fluxes in Europe"

Line 83 "so that changes in soil-water storage over the period can be neglected" -> this is incorrect. The fact that soil moisture anomalies are small at the beginning and end of drought events does not say anything about actual soil moisture.

A4: We have clarified this sentence in lines 93-95:

"Soil moisture levels are therefore close to normal (i.e., unchanged by drought and thus zero anomaly) at the beginning and end of each drought period, such that changes in soil-water storage beyond the mean seasonal variations over the period can be neglected."

Line 101-104 -> Yes, but this is only true if all fluxes are accounted for (including deeper flow over catchment boundaries), and of course when using independently observations of all balance components the balance will generally not close because of measurement uncertainties. Note that only precipitation is measured as a mass-flux in rain gauges; discharge is measured per volume, and evaporation often as energy flux.

A5: We have clarified this sentence in lines 119-122:

"By fundamental water balance, any net imbalance of input (precipitation) and output (ET, runoff) fluxes occurring at any point in time during the drought period must correspond to an associated change in water storage within the catchment at that time (assuming negligible roles of measurement uncertainties and groundwater flow over catchment boundaries)."

In addition, in lines 76-78 we added the information that all flux variables are converted to the common unit of mm/day before the analysis:

"Further, all considered fluxes are normalized by catchment area to units of mm/day prior to the analysis."

Line 120 "(state of the art)" -> One could also argue that using a "bucket-style" model forced by Priestley-Taylor potential ET is not really state-of-the-art, but I agree it is one of the more reliable products available.

A6: We have replaced the term "state-of-the-art" by "commonly used" in line 137.

Reviewer #2

The authors have substantially revised the manuscript and have made it thereby significantly easier to grasp the details and the extent of their analysis.

B1: We thank the reviewer for acknowledging our revisions.

While this clearly helps to understand how the conclusions are supported by the results, it also reveals weak-points of their assessment which question originality of the work as well as the robustness of their conclusions.

Issue 1:

Originality of the analysis

Large aspects of the analysis are equivalent to the analysis conducted by Nicolai-Shaw et al. (2017, doi: 10.1016/j.rse.2017.06.014), which also produce a set of figures similar to Figure 2 (and versions thereof). In particular, Figure S2 (supporting information) of Nicolai-Shaw et al. (2017) features drought response composites for P, E and NDVI for the three European macro regions. This figure also shows several features highlighted by Orth & Destouni, including e.g. the delayed response of ET and NDVI as well as the positive NDVI anomalies prior to the strongest soil moisture anomaly.

This leaves the impression that Orth & Destouni are trying to hide issues with the originality of their analysis.

I Acknowledge that the analysis of runoff and the interpretation related to that is substantially novel.

B2: We have now clarified further that we use the drought composite approach introduced by Nicolai-Shaw et al. (2017), early in the manuscript, in lines 79-81:

“In order to comprehensively and for the first time investigate the partitioning of drought-related water deficits between blue and green water fluxes, we employ a drought composite analysis approach (27).”

In addition, we have also clarified that we move beyond this drought composite approach by analysing integrated drought anomalies (e.g. in Figures 3 and 4) in lines 98-100:

“Moving beyond the drought composite analysis in (27), we furthermore quantify accumulated anomalies of precipitation, runoff and ET, and refer to these in following as the drought-related anomalies of each flux variable.”

Moreover, as recognized also by the reviewer, a key difference between our study and the Nicolai-Shaw et al. paper is that runoff analysis is totally missing there, while it is a key focal issue in our paper. By neglecting runoff, their paper does not actually address the partitioning and evolution of drought anomalies among and through all hydrological balance components: ET, runoff and soil-moisture change, which is a main aim of our study. This is clearly stated in various prominent places in across the manuscript, such as for example in the abstract, introduction, and conclusions.

Further, our actual results, with generally stronger and faster runoff than ET effects of drought, clearly show that the Nicolai-Shaw 2017 study has then actually omitted the most important part of the hydrological drought-impact partitioning and evolution, i.e., the runoff part. Our results thus point out as incomplete and, as such, biased and potentially misleading, the common approach, as in the Nicolai-Shaw et al. 2017 paper, to just cut off and only analyse the uppermost part of the soil in assessments of hydrological anomaly/variability/change evolution and impact spreading through the landscape and catchments.

Issue 2:

Incomplete use of considered data sets suggests either a sloppy approach or that the authors are hiding unwanted results.

This severely questions the robustness of their conclusions.

Analysing water fluxes at continental scales from an observational perspective will always be a compromise between using sparse station observations and model-based estimates which provide full spatial and temporal coverage. I, therefore, acknowledge the authors efforts to base their assessment on both station observations and model-based estimates.

B3: We agree with the reviewer. See also response B5.

However, the authors are incomplete with their assessment, raising questions regarding the reliability of the conclusions.

Specifically.

(2.a) Using only 3 of dozens of FLUXNET stations raises the question what happens if other stations are considered. I suspect that some of the results might vanish, questioning the conclusions drawn by the authors.

B4: In response to the reviewers concern we have updated Figure S2 to triple the number of considered Fluxnet sites. The result is the same as in the previous version of that Figure – it highlights the usefulness of the employed gridded continent GLEAM ET dataset by showing that it agrees reasonably well with ground-truth measurements, especially in comparison with other state-of-the-art model datasets.

Please note that we cannot employ “dozens” of Fluxnet stations in this analysis as they (i) do not provide sufficiently long time series (~10 years) in order to capture enough droughts to compute meaningful composites, and as (ii) they are not distributed equally across Europe, such that there are only few stations in northern and southern Europe.

(2.b) Partial use of model estimates. The authors use model estimates of soil moisture and evapotranspiration. But why is modelled runoff not considered to augment the analysis of the runoff observations. To me this is inconsequential – if evapotranspiration and soil moisture are considered reliable enough to draw strong conclusions, the analysis should be extended to runoff simulations of all models. Again, this raises the question whether the conclusions remain robust if the full data set is considered.

B5: We have re-structured and expanded the corresponding paragraph in the manuscript to motivate the selection of datasets chosen in our study (lines 62-71):

“Analysing continental water fluxes from an observational perspective, as is the aim of this study, is necessarily a compromise between using sparse station observations and model-based estimates that provide full spatial and temporal coverage. Addressing this issue, we employ runoff measurements from catchments distributed across Europe (23, see methods) and gridded precipitation data derived by upscaling station observations (24), along with gridded reanalysis-like data on ET (25) and soil moisture (26). While the latter products are model-based, they are validated against corresponding station observations (25, 26); to further check their usefulness for our analysis we compare them against more

station observations, and repeat the analysis by replacing them with similar datasets obtained with different models to ensure negligible influence of particular models on our conclusions (see below).”

Within that paragraph we have incorporated an adapted version of a statement made by the reviewer above our response B3:

“Analysing water fluxes at continental scales from an observational perspective will always be a compromise between using sparse station observations and model-based estimates which provide full spatial and temporal coverage.”

We fully agree with that. While we aim to use observations to the highest degree possible in this study, we cannot avoid model-derived products in some cases (namely for soil moisture and ET). In these cases, however, we use established products, and validate them even further against existing station observations. In the other cases, where we already use observation(-based) data (namely for precipitation and runoff), we do not think it would add insights to the study to employ also model-based products as suggested by the reviewer. This would even contradict the initially stated goal of employing observations wherever possible.

(2.c) Very small and biased model ensemble. The main analysis is based on three model products (SWBM (soil-moisture), ERA-Interim/Land (soil-moisture, ET), GLEAM (ET)), which are used for different purposes of the analysis. Generally, ERA-Interim/Land is used to demonstrate the robustness of the results derived from the other products. However, considering the availability of comprehensive global hydrological model ensembles (see e.g. <http://www.earth2observe.eu/>, or ISIMIP2a doi:10.5880/PIK.2017.010) this choice appears to be limited and biased towards the authors own work (SWBM is a product developed by R. Orth). Given the large uncertainties involved in large scale hydrological modelling this raises questions w.r.t the robustness of the presented assessment which can only be resolved if a larger model ensemble is considered.

B6: In order to underline the usefulness of the soil moisture dataset for this study we have validated the dataset against independent station observations in the new Figure S3, as done previously, and now further extended (see B4), for ET in Figure S2.

We mention this additional validation in the manuscript in lines 139-143:

“Furthermore, there is significant agreement between the employed soil moisture product and independent measurements across the considered climate regimes (Figure S3). This suggests that drought periods are correctly captured by the soil moisture product, which is further indicated by the consistent observed runoff and precipitation responses shown in Figure 2.”

Moreover, we do not agree with the reviewer in terms of a model ensemble.

Our main finding in this study, the stronger and faster drought impacts on blue-water runoff than on green-water ET, is strong and robust as runoff is clearly (much) more responding to droughts than ET. We have obtained this finding, and highlighted its robustness, with a necessary and meaningful combination of observation-based and model-based datasets, as explained in B5.

In this context, we feel that it is sufficient to use the described suite of datasets, comprising gridded upscaled observations, station observations, and established datasets from three independent models. Given the variety of data already considered, we do not think that exploration of further model data can substantially add to our analysis. Further, the inclusion of further model-based datasets from models with potentially insufficient representations of the land hydrology and biosphere, especially in the case of extreme events such as droughts, could even lead to inaccurate or false results, as shown with the MERRA dataset in Figure S2.

Issue 3

Over-interpretation of some of the results (examples, to illustrate the tendency

While reading the paper, I repeatedly have the impression that the authors tend to over-interpret their results, which sometimes results in erroneous conclusions

Two examples:

(3.a) The authors discuss what they refer to as “blocking of the water cycle”, and describe one plausible mechanism. However, what they observe is essentially a non-linear response of runoff to storage which in turn is common knowledge in catchment sciences (see e.g.: Kirchner (2009, doi: 10.1029/2008WR006912) or Clark et al (2011, doi: 10.1002/hyp.7902).

B7: We have included the references mentioned by the reviewer to clarify that this has been observed in previous studies (lines 126-127):

“This illustrates a blocking of the water cycle, and is in line with previous studies (28, 29).”

(3.b) In Figure 3b, they state “drought promotes biosphere”, which is plainly wrong. What the figure shows is the fact that in humid climates good weather periods foster vegetation growths. Water limitations only play a role for extremely low absolute soil moisture values. The fact that the authors hint at this mechanism in the text is acknowledged but overshadowed by the misleading statement in the figure.

B8: We have clarified this point in the manuscript in lines 202-207:

“The drought-induced ET and NDVI increases in humid areas may seem counter-intuitive; however, they can be explained as droughts are not only characterised by reduced water availability, but also by increased radiation and hence energy availability (Figure S4). In that sense, drought in wet and therefore energy-limited climate regimes (e.g. Norway) promotes biospheric activity, as seen from increases in ET and NDVI, because this is primarily controlled by (increased) net radiation.”

Given this comprehensive explanation of the ET increases in Figure 3b (which in its previous version was also acknowledged by the reviewer), and given the mentioning of our drought definition in several places across the manuscript, we decided to keep the short statements in Figure 3b.

Issue 4

Some of the graphics are non-transparent, misleading or wrong

Examples

(4.a) Figure 2 remains to have a non-logical x-axis which includes only positive values but goes through zero. This leads to strange plotting behaviour with a variable starting “positive” crossing zero and going again to “positive” but on the other side of the plot. Even after repeated thinking and the authors explanation in the rebuttal I could not get my head around that. The fact that this non-physical axis is even not mentioned in the caption does not make this better, and will contribute to confusing any potential reader of this assessment.

B9: Reporting the actual anomalies would make it difficult to keep an overview on what is building-up/intensifying drought and what is recovering/dampening drought – for example, negative precipitation anomalies are strengthening drought, while negative runoff and ET anomalies are dampening drought. To

prevent this confusion, we have decided to ignore the actual sign of the anomalies and instead group them in terms of whether they build-up /intensify drought or recover/dampen drought.

We have clarified this point in the manuscript directly after the figure is introduced in lines 103-105:

“Figure 2 shows the mean temporal evolution of precipitation, runoff and ET anomalies during drought, with the anomalies grouped in terms of whether they intensify or dampen drought, instead of their actual sign.”

(4.a) Figure 4 why are all values positive, even when the values in Fig 2 do cross the zero line? To me this remains un-accessible and I doubt that it will be accessible to most of the readers.

B10: See B9. Also for this figure we have clarified our handling of the anomalies in the manuscript directly after the figure is introduced in lines 224-225:

“As in Figure 2, anomalies are not shown with their actual sign, but grouped in terms of whether they build up or recover drought.”

Reviewer #3

I am happy that the authors have addressed my comments from the first review, though I hold firm that the title needs changing.

I think it is important that the title reflects what can be concluded from the paper. I understand the authors' comment about the word limit, but I believe that the maximum word limit for the title is 15 words, therefore the following would be fine:

"European drought impacts are climate-driven for green while always strong for blue water fluxes"

C1: We have changed the title in order to reflect the European focus of our study.

“Drought reduces blue much stronger than green water fluxes in Europe”

A Europe-wide analysis is still a significant achievement that is worthy for publication in Nature Communications, there is no need to overlay the results.

C2: We thank the reviewer for underlining the significance of our results.

Reviewer #2 (Remarks to the Author):

Overall the authors have addressed most of my remaining questions, e.g. by (i) providing an assessment of additional flux net stations (and having a transparent station inclusion criterion) and by (ii) explaining their reasoning for only including a limited (and biased) sub-set of model estimates to augment sparse observations.

Nevertheless, the following issues remain:

(1) To me it is an open question, whether the results stay stable if a larger (well-behaved) model ensemble is considered.

(2) Response "B7" cites studies mentioned in the previous review and uses them to refer to a "blocking of the hydrological cycle". However, these studies report general non-linear rainfall-runoff behaviour which does not equate to a "blocking". Therefore, I find that these citations are not considered accurately.

(3) In my view the statement "drought promotes biosphere" in Figure 3 is still misleading, as it is not the unusually dry conditions which trigger plant activity but a confounding effect. Adjusting the drought definitions seems to me counter-productive as it reduces the clarity of the presented results. I acknowledge that drought definitions are difficult and depend on context, but the term "drought" is commonly exclusively associated with a water deficit.

(4) I find the axis in Figure 2 (and partially also 4) still not logical, as it does not make sense to have number line (https://en.wikipedia.org/wiki/Number_line) go through zero without changing the sign. Instead I wonder, why the authors did e.g. not simply plot sth. Like "minus Precipitation" to ensure that all variables point in the same direction.

(5) Small mistake in the caption of Fig S3: "... ET. (b)-(d) Comparison ..." should read "ET. (b)-(j) Comparison"

Finally, I acknowledge that some of the points mentioned above may either go beyond the scope of the presented study (1), are subject to interpretation (3), or depend on perception (4). Therefore, I would like to leave a final evaluation of these points to the editor.

We want to especially thank reviewer #2 for the continued and fruitful discussion of our manuscript. We have implemented most of the suggested changes from the final review, as explained below:

Overall the authors have addressed most of my remaining questions, e.g. by (i) providing an assessment of additional flux net stations (and having a transparent station inclusion criterion) and by (ii) explaining their reasoning for only including a limited (and biased) sub-set of model estimates to augment sparse observations.

A1: We thank the reviewer for acknowledging our revisions.

Nevertheless, the following issues remain:

(1) To me it is an open question, whether the results stay stable if a larger (well-behaved) model ensemble is considered.

A2: We feel that this is beyond the scope of the present study, as also acknowledged by the reviewer below.

(2) Response “B7” cites studies mentioned in the previous review and uses them to refer to a “blocking of the hydrological cycle”. However, these studies report general non-linear rainfall-runoff behaviour which does not equate to a “blocking”. Therefore, I find that these citations are not considered accurately.

A3: We have updated the sentence in lines 132-133:

“This illustrates a blocking of the water cycle. Similar non-linear rainfall-runoff behavior has been found in previous studies (28, 29).”

(3) In my view the statement “drought promotes biosphere” in Figure 3 is still misleading, as it is not the unusually dry conditions which trigger plant activity but a confounding effect. Adjusting the drought definitions seems to me counter-productive as it reduces the clarity of the presented results. I acknowledge that drought definitions are difficult and depend on context, but the term “drought” is commonly exclusively associated with a water deficit.

A4: We have removed the explanatory text from Figure 3.

(4) I find the axis in Figure 2 (and partially also 4) still not logical, as it does not make sense to have number line (https://en.wikipedia.org/wiki/Number_line) go through zero without changing the sign. Instead I wonder, why the authors did e.g. not simply plot sth. Like “minus Precipitation” to ensure that all variables point in the same direction.

A5: We have implemented the reviewer’s suggestion in Figure 2, and Supplementary Figures 1 and 4.

(5) Small mistake in the caption of Fig S3: "... ET. (b)-(d) Comparison ..." should read "ET. (b)-(j) Comparison"

A6: We thank the reviewer for spotting this, and have corrected the caption of Supplementary Figure 2 accordingly.

Finally, I acknowledge that some of the points mentioned above may either go beyond the scope of the presented study (1), are subject to interpretation (3), or depend on perception (4). Therefore, I would like to leave a final evaluation of these points to the editor.